# Early and Long-Term Outcomes after Propofol-and Sevoflurane-Based Anesthesia in Colorectal Cancer Surgery: A Retrospective Study

**DOI:** 10.3390/jcm11092648

**Published:** 2022-05-08

**Authors:** Seungwon Lee, Dae Hee Pyo, Woo Seog Sim, Woo Young Lee, MiHye Park

**Affiliations:** 1Department of Anesthesiology and Pain Medicine, Samsung Medical Center, Sungkyunkwan University School of Medicine, 81 Irwon-ro, Gangnam-gu, Seoul 06351, Korea; seungwon0209.lee@samsung.com (S.L.); wooseog.sim@samsung.com (W.S.S.); 2Department of Surgery, Samsung Medical Center, Sungkyunkwan University School of Medicine, Seoul 06351, Korea; daehee.pyo@samsung.com

**Keywords:** colorectal cancer, cancer resection, cancer recurrence, general anesthesia, lymphocyte, neutrophil, propofol, sevoflurane

## Abstract

Background: Propofol is considered to protect against immunosuppression and has lower inflammatory responses in the perioperative period than volatile agents. We evaluated whether the anesthetic agent is associated with cancer outcomes. Methods: We retrospectively reviewed 2616 patients who underwent colorectal cancer surgery under general anesthesia between 2016 and 2018 (follow-up closure: July 2021) at a single institution. Patients received propofol-based total intravenous anesthesia or sevoflurane-based inhalational anesthesia. After propensity score matching, the postoperative neutrophil-lymphocyte ratio (NLR) was compared as primary outcome, and clinical outcomes were evaluated. Results: After 1:2 propensity matching, 717 patients were given propofol anesthesia and 1410 patients were given sevoflurane anesthesia. In the matched cohort, preoperative NLR was not significantly different between propofol and sevoflurane anesthesia (mean (95% CI)2.3 (1.8 to 2.8) and 2.2 (1.9 to 3.2); *p* = 0.72). NLR was significantly lower in propofol anesthesia at postoperative day two and five (mean difference (95% CI) 0.71 (0.43 to 0.98); *p* = 0.000 and 0.52 (0.30 to 0.74); *p* = 0.000). Urinary retention showed a higher incidence after propofol anesthesia (4.9% vs. 2.6%; *p* = 0.008). Other postoperative complications and overall/recurrence-free survival were not different in the two groups. Discussion: Although propofol anesthesia showed lower postoperative NLR than sevoflurane anesthesia, there was no association with clinical outcomes.

## 1. Introduction

Cancers of the colorectum, breast, prostate, and lung represent almost half of the overall cancer burden worldwide [1]. Surgical resection remains a mainstay of treatment for long-term survival in these solid cancers. However, if minimal residual micrometastases are not eliminated by the immune system or proliferate by systemic response, surgery can be a chance for metastatic disease [2,3]. Thus, surgical stress, inflammation, and the host’s immune system are important factors that affect the outcomes of cancer.

Previous studies have shown that volatile inhalational agents alter immune processes and are proinflammatory, appearing to increase the incidence of cancer metastases in mice and humans [4]. In contrast, propofol appears to suppress tumor growth and reduce the risk of metastases in mice and humans because of its anti-inflammatory and antioxidative activities [5,6]. Recent studies have indicated the increased recurrence or mortality in colorectal cancer patients undergoing inhalational anesthesia compared with total intravenous anesthesia [7,8].

The neutrophil-lymphocyte ratio (NLR) has been suggested to be a simple index of systemic inflammatory response [9]. Neutrophilia occurs during systemic inflammation, and lymphopenia is a marker for depressed cell-mediated immunity. There have been a few studies to assess the influence of anesthetic agents on changes in inflammatory markers within the postoperative period and to clarify the impact of postoperative alterations on the survival of patients with cancer [10,11].

Therefore, we hypothesized that patients receiving propofol-based total intravenous anesthesia would have a superior inflammatory response and thereby less tumor recurrence compared to sevoflurane-based inhalational anesthesia after colorectal cancer surgery. The primary endpoint was NLR, which were compared between the two groups that had received propofol or sevoflurane as the main anesthetic agent. Secondary outcomes were early postoperative complications and recurrence-free survival and overall survival.

## 2. Materials and Method

### 2.1. Study Population and Data Collection

Our institution operates as a paperless hospital with an electronic medical record system that archives all patient medical information. This retrospective study was approved by the Samsung Medical Center Institutional Review Board (IRB No. SMC 2021-08-004). As all data in this study were curated using “Clinical Data Warehouse Darwin-C,” an electronic system designed to search and retrieve de-identified medical records, individual consent was waived by the Institutional Review Board. The study was conducted in accordance with the principles of the Declaration of Helsinki.

The study population consisted of patients over 18 years of age who underwent primary colorectal cancer surgery under general anesthesia between December 2016 and December 2018 at the Samsung Medical Center. The exclusion criteria were other inhalation anesthesia (desflurane or isoflurane) or incomplete data collection.

Demographic and clinical data were extracted from the Clinical Data Warehouse Darwin-C of Samsung Medical Center. As baseline patient characteristics, we collected information on age, sex, American Society of Anesthesiologists (ASA) physical status, smoking history, alcohol consumption, previous abdominal surgical history, the presence of underlying disease (hypertension, diabetes mellitus, stroke, chronic renal disease, coronary artery disease, heart failure, and chronic obstructive pulmonary disease) and the results of preoperative laboratory tests (hemoglobin, albumin, neutrophil, lymphocyte). We extracted the following intraoperative and surgical data: duration of anesthesia, anesthetic agents used for maintenance of general anesthesia, intraoperative transfusion, operation type, tumor location, stage, invasion). NLR was calculated as the absolute count of neutrophils (number/µL) divided by the absolute count of lymphocytes (number/µL). For postoperative data, we collected adjuvant treatment and early postoperative complications during hospitalization (wound problem; occurrence of infection involving the skin or subcutaneous tissue and requiring surgical re-intervention, anastomosis site leakage; diagnosed by radiographic findings, ileus; symptomatic and diagnosed by radiographic findings, intraabdominal fluid/abscess; diagnosed by radiographic findings, sepsis; despite adequate fluid resuscitation, patients have hypotension requiring vasopressors to maintain a mean arterial blood pressure above 65 mm Hg and have an elevated serum lactate concentration of more than 2 mmol/L resulting from dysregulated host responses to infection, myocardiac infarction; detection of a rise of cardiac troponin values with symptoms, pulmonary complications; requiring treatment with antibiotics for a suspected respiratory infection or management by respiratory care physiotherapists for lung care, cerebral infarction; diagnosed by imagining findings, vascular complication; diagnosed by imagining findings, urinary retention; need for in-and-out catheterization or reinsertion of an indwelling urinary catheter during the hospital stay after the original urinary catheter had been removed, re-operation).

### 2.2. Anesthesia

Patients were assigned to the propofol group or the sevoflurane group according to the type of anesthesia received, which was based on the anesthesiologist’s preference. In the propofol group, anesthesia was induced and maintained via the target-controlled infusion of propofol and remifentanil intravenously. Patients in the sevoflurane group received sevoflurane inhalational anesthesia and a supplementary intravenous opioid at the discretion of the anesthesiologist. Intravenous hydromorphone was given at the end of surgery for early postoperative management of analgesia. Patients received a patient controlled analgesia with fentanyl, programmed to deliver a bolus of 15 µg on demand with a lockout period of 15 min. These analgesic regimens were applied for approximately 72 h postoperatively. No regional anesthesia was adjusted in patients who underwent colorectal cancer surgery according to our institution protocol.

### 2.3. Postoperative Follow-Up

The patients were followed up postoperatively in the outpatient clinic every three months for the first two years, and every six months for the next three years, and then annually thereafter. At every visit, interim clinical history, laboratory tests including complete blood count, inflammatory markers, and liver enzyme and chest radiograph were checked. The serum levels of carcinoembryonic antigen and the computed tomography of the abdominopelvic/chest area were evaluated every six months. Colonoscopy was performed at the postoperative first year and then biennially.

### 2.4. Statistical Analysis

Survival time was defined as the interval between the date of surgery and death, or 15 July 2021 for those who were censored. All data are presented as mean ± (standard deviation, SD), median (interquantile, IQR), or number (percentage). For other discrete variables, proportions of patients between groups were compared with a Chi-square test or Fisher’s exact test. For continuous variables, between-group differences were assessed with a Student’s t-test or a Mann-Whitey U-test according to the normality of the data.

To account for differences in baseline characteristics and perioperative potential confounding factors, propensity score analysis was used to account for intergroup differences according to the anesthetic agent. All demographic and perioperative parameters displayed in Table 1 were used in the adjustment with propensity score analysis. Pre-specified outcomes were the postoperative NLR, postoperative complications, recurrence-free survival and overall survival. After propensity matching, NLR as primary outcome was compared with logistic regression using generalized estimating equations and adjusted with Bonferroni correction.

The Kaplan-Meier method was used to calculate the overall survival of patients from the date of surgery to the date of death; patients alive were censored from the follow-up closure data (15 July 2021). The recurrence-free survival and overall survival were compared with the Cox proportional hazard model after propensity score matching. Univariate and multivariable Cox-regression analyses were performed to identify independent risk factors of recurrence and mortality in overall patients. Statistical analyses were performed using SPSS 27.0 (IBM Corp., Chicago, IL, USA) or R 4.0.2 (R Development Core Team, Vienna, Austria; http://www.R-project.org/ (June 2020)). *p* < 0.05 was considered indicative of statistical significance.

Patients who received elective colorectal cancer surgery between December 2016 and December 2018 were included. We hypothesized that those patients receiving propofol anesthesia would have lower NLR compared to those undergoing sevoflurane anesthesia after colorectal surgery at postoperative day five. Based our previous study, we assumed the difference of NLR to be 0.5 with an SD of 2.5 [9]. To achieve a power of 90% and a two-tailed type I error rate of α = 0.05, of which 30% are in the propofol group and 70% are in the sevoflurane group, 419 patients in the propofol group and 964 patients in the sevoflurane group were needed in unmatched groups.

## 3. Results

In total, 2616 patients underwent primary colorectal cancer surgery from December 2016 to December 2018 at Samsung Medical Center and 2571 patients satisfied the selection criteria. Of these, 1852 (72%) received sevoflurane-based general anesthesia and 719 (28%) received propofol-based general anesthesia (Figure 1). Most baseline and operative characteristics of the study population were well-balanced between the two groups, except type of operation; the proportion of robotic surgical procedure was significantly higher in the sevoflurane group (10.2% vs. 0.1%). Most patients of robotic surgery belonged to the sevoflurane group because the choice of agent was at the discretion of each attending anesthesiologist. The median follow-up period was 42.4 (95% CI, 41.9 to 42.6) months for all patients, 42.1 (95% CI, 41.8 to 43.0) months for the propofol group, and 42.5 (95% CI, 41.8 to 42.6) months for the sevoflurane group. After 1:2 propensity score matching, 717 pairs were matched (717 patients remained in the propofol group and 1410 patients remained in the sevoflurane group). Table 1 shows the characteristics and intraoperative variables for the total study cohort and those for the propensity matched cohort. All standardized mean differences for the study variables were less than 0.1 in the propensity matched cohort.

### 3.1. Change of Neurtophil-Lymphocyte Ratio and Early Postoperative Complications

Neutrophil-lymphocyte data were available in 2127 patients at preoperative exam, 1382 patients at postoperative day one, 2001 patients at postoperative day two, 2071 patients at postoperative day five and 2061 patients at postoperative day 20–30 at the first outpatient clinic visit. Before and after propensity score matching, the change of NLR was shown in the Figure 2. Preoperative NLR was not significantly different between the propofol group 2.3 (95% CI 1.8 to 2.8) and the sevoflurane group 2.2 (95% CI 1.9 to 3.2) (*p* = 0.72) in the matched cohort. The NLR exhibited significant increases compared with the baseline until day five postoperative, independent of anesthetic technique. NLR at postoperative day two (mean difference (95% CI) 0.71 (0.43 to 0.98); *p* = 0.000) and five (mean difference (95% CI) 0.52 (0.30 to 0.74); *p* = 0.000) were significantly lower after propofol anesthesia than sevoflurane anesthesia. There were no differences in early postoperative complications except urinary retention (Table 2). Patients who complained of urinary retention were more prevalent in the propofol group (36/717 (4.9%) vs. 35/1410 (2.6%); *p* = 0.008).

### 3.2. Recurrence-Free Survival and Overall Survival

The Kaplan-Meier survival curves demonstrated 53-month recurrence-free survival rates of 86.4% in the propofol group and 85.6% in the sevoflurane group, and 53-month overall survival rates of 97.0% and 96.1%, respectively. The multivariable Cox regression analysis in overall patients demonstrated no significant association between anesthesia agent and recurrence-free survival (hazard ratio, 1.04; 95% CI, 0.80 to 1.34; *p* = 0.80) and overall survival (hazard ratio, 1.12; 95% CI, 1.06 to 1.17; *p* = 0.91) (Table 3).

After propensity score matching, recurrence-free survival (hazard ratio, 1.08; 95% CI, 0.84 to 1.40; *p* = 0.54) and overall survival (hazard ratio, 1.09; 95% CI, 0.68 to 1.75; *p* = 0.72) did not significantly differ between the two groups regardless of sevoflurane or propofol usage. Figure 3 shows the Cox proportional hazard model for overall survival and recurrence-free survival after propensity score matching.

## 4. Discussion

In this retrospective study of patients undergoing colorectal cancer surgery, we evaluated the effect of anesthetic agents on the NLR and clinical outcomes. Although the NLR at postoperative day two and five were significantly lower with propofol anesthesia, this benefit did not translate into a clinically significant reduction in the occurrence of early postoperative complications including infections or long-term prognosis.

Previous large studies including breast, gastric, liver, and colorectal cancer have indicated increased mortality in cancer patients undergoing inhalation anesthesia [12,13]. Nonetheless, the association with mortality found in these studies might not be caused by cancer recurrence. In studies of colorectal cancer surgery, the anesthetic techniques were not standardized, and results from several studies remain mixed [7,8,14]. A single center trial demonstrated that propofol-based total intravenous anesthesia in colon cancer surgery was associated with better survival than desflurane anesthesia [7]. However, a recent nationwide registry-based cohort study of 4347 individuals in each of the inhalational and total intravenous anesthesia groups with balanced baseline covariates found a weak association between recurrence and exposure to inhalational anesthesia, but no association for all-cause mortality or disease-free survival [8]. However, neither study considered the duration of anesthetic agent exposure and potentially limiting data for comparing the influence on postoperative inflammation or immunity from anesthetic agents.

Propofol and sevoflurane were the most widely used anesthetic agents. Propofol is considered to protect against immunosuppression during the perioperative period and has a lower inflammatory response than volatile agents [15,16]. Furthermore, propofol is reported to have anti-inflammatory properties targeting neutrophil activity [17,18]. Conversely, sevoflurane has been reported to suppress the immune response by regulating cytokine expression and reducing natural killer cell toxicity. In addition, sevoflurane can induce tumor stem cell proliferation and the expression of oncogenic protein markers [19].

Neutrophils play a critical role in tumorigenesis, progression, and metastasis in multiple ways, including both direct effects on cancer cells and indirect effects on the tumor microenvironment [20]. High tumor infiltration and lymphocyte densities were associated with improved survival outcomes. The apoptosis of tumor infiltrating lymphocytes mediates resistance to cancer immunotherapy [21]. Increased preoperative NLR is associated with lymph node metastasis and distant metastasis, as well as treatment resistance [22]. Increased preoperative NLR is a recognized inflammatory marker for poor prognosis in solid tumors and colorectal cancer [22,23,24,25]. Previous studies that compared the propofol/paravertebral block and sevoflurane/opioid anesthesia showed a reduction in NLR in propofol anesthesia, but the primary perioperative analgesic technique significantly affected that result [9]. Regional anesthesia imparts less surgical stress and has short-term advantages, because the cellular immune response appears to be less affected by reduced pain [26,27,28,29].

Consistent with previous reports, our multivariable analysis showed that preoperative NLR was associated with recurrence and death. Our findings also demonstrated that the NLR at postoperative days two and five was significantly lower in propofol anesthesia. However, the difference was lost when the observation period was extended to postoperative day 20–30. Recently, the association of immediate postoperative inflammation and tumor prognosis has been of interest [11,30,31]. However, the timing of postoperative blood tests was not consistent in those studies. Also, the prognostic value of NLR in the early postoperative period might be limited, because surgical stress and wound healing have an impact on the inflammatory indicators [32]. There was still a paucity of data because of the small sample size studies or the limited reporting on postoperative NLR. The effect of postoperative NLR in cancer outcomes could not be explained in this study because we did not compare long-term outcomes according to postoperative NLR. Further research on the clinical meaning of increased postoperative NLR in cancer patients is needed.

We found that the difference of postoperative early NLR was not associated with meaningful clinical outcomes in early postoperative complications. However, we found an interesting effect on urinary retention in the two anesthetic techniques. The rate of postoperative urinary retention in colon resection is about 2% and up to 24% for rectal resection [33]. Risk factors for prolonged urinary retention after colorectal surgery include older age, male sex, longer operative time, pelvic dissection, and low rectal cancers [34]. Previous studies have frequently focused on the effects of sevoflurane and propofol anesthesia on postoperative pain, cognitive dysfunction, or nausea/vomiting. Thus, little evidence has been demonstrated for a clinically significant association between total intravenous anesthesia and urinary retention, regardless of surgical type. However, urinary retention is a common complication after general anesthesia and reduces the quality of recovery. Further investigations are needed to determine the effect of anesthetic agents on urinary retention.

This study has several limitations for generalizability because of its inherent retrospective observational design. First, although we collected as much information as possible on confounding factors affecting mortality and infection, there might be additional unknown variables not included in the study. It was not possible to compare total amounts of opioids in the two groups. Perioperative factors such as maintaining normothermia or providing supplemental oxygen influencing immune competence might modulate the risk of recurrence or metastasis. Second, this study included patients from a single institution. However, it has several merits. The study population was relatively large and the treatment strategies did not change during the study, reducing selection bias. Third, the median follow-up was less than five years. Importantly, the 53-month survival rate was greater 96%; thus, the 50-month follow-up time in our study might not be sufficient to detect a meaningful difference of the effect of anesthesia type on survival. However, previous studies showed 73% survival of all recurrences within the first 24 months postoperatively [8,35].

Although the NLR at postoperative day two and five was lower after propofol-based total intravenous anesthesia compared to sevoflurane-based inhalational anesthesia, this benefit did not translate into a clinically significant reduction in the occurrence of early postoperative complications or long-term survival in patients who underwent colorectal cancer surgery. Further large-scale, multicenter studies are required to confirm the definite conclusion of the anesthetic factors and cancer outcomes. Further studies focusing on the difference in postoperative NLR values according to the anesthetic technique revealed in this study affects long-term prognosis are needed.

## Figures and Tables

**Figure 1 jcm-11-02648-f001:**
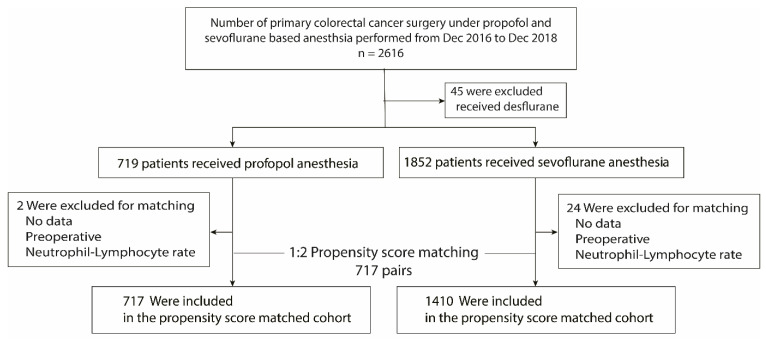
Flow diagram detailing the selection of patients included in the retrospective analysis.

**Figure 2 jcm-11-02648-f002:**
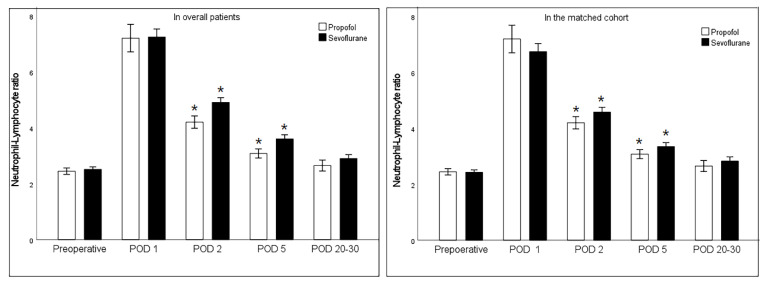
Change of neutrophil-lymphocyte ratio in the perioperative period. POD; postoperative day. * means *p* < 0.05 when compared with logistic regression using generalized estimating equations with Bonferroni correction in a propensity score matched cohort.

**Figure 3 jcm-11-02648-f003:**
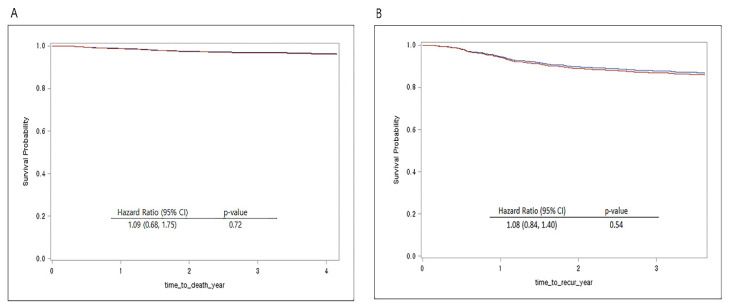
(**A**) Overall survival curves from the date of surgery by anesthesia type in the propensity score matched cohort. (**B**) Recurrence-free survival curves from the date of surgery by anesthesia type in the propensity score matched cohort. (red line: propofol received, blue line: sevoflurane received).

**Table 1 jcm-11-02648-t001:** Baseline characteristics of study population before and after propensity score matching.

	Overall Patients			After Matching		
	Propofol	Sevoflurane	SMD	Propofol	Sevoflurane	SMD
	(*n* = 719)	(*n* = 1852)		(*n* = 717)	(*n* = 1410)	
Age, year	61.7 (11.4)	60.7 (12.0)	0.084	61.7 (11.4)	61.8 (11.8)	0.008
Sex, female	290 (40.3)	797 (43.0)	0.055	289 (40.3)	597 (42.3)	0.041
BMI, kg/m^2^	24.1 (3.3)	23.8 (3.4)	0.078	24.1 (3.3)	23.9 (3.4)	0.035
ASA						
I	179 (24.9)	539 (29.1)	0.094	178 (24.8)	356 (25.3)	0.009
II	485 (67.5)	1139 (61.5)	0.124	484 (67.5)	934 (66.2)	0.026
III	55 (7.7)	166 (9.0)	0.047	55 (7.67)	120 (8.5)	0.031
IV	0 (0)	8 (0.4)		0 (0)	0 (0)	
Current smoking	70 (9.7)	197 (10.6)	0.029	70 (9.8)	140 (9.9)	0.005
Heavy drinking	58 (8.1)	129 (7.0)	0.042	58 (8.1)	103 (7.3)	0.03
History of surgery	169 (23.5)	451 (24.4)	0.019	168 (23.4)	343 (24.3)	0.021
Comorbidities						
Hypertension	271 (37.7)	633 (34.2)	0.074	271 (37.8)	512 (36.3)	0.031
Diabetes mellitus	149 (20.7)	321 (17.3)	0.087	148 (20.6)	273 (19.4)	0.032
Stroke	18 (2.5)	59 (3.2)	0.041	18 (2.5)	31 (2.2)	0.021
CAD	26 (3.6)	85 (4.6)	0.049	26 (3.6)	53 (3.8)	0.007
Heart failure	1 (0.1)	16 (0.9)	0.103	1 (0.1)	2 (0.1)	0.001
COPD	18 (2.5)	54 (2.9)	0.028	18 (2.5)	34 (2.4)	0.007
Preoperative test						
Haemoglobin, g/dL	13.4 [12.0, 14.5]	13.0 [11.6, 14.3]	0.071	13.4 [12.0, 14.5]	13.0 [11.6, 14.4]	0.074
Albumin, g/dL	4.4 [4.2, 4.7]	4.4 [4.2, 4.6]	0.039	4.4 [4.2, 4.7]	4.4 [4.2, 4.6]	0.024
Creatinine, mg/dL	0.81 [0.68, 0.95]	0.80 [0.68, 0.94]	0.021	0.81 [0.68, 0.95]	0.81 [0.69, 0.94]	0.005
NLR	2.1 [1.5, 2.9]	2.1 [1.5, 2.9]	0.037	2.1 [1.5, 2.9]	2.0 [1.5, 2.8]	0.012
Neoadjuvant therapy	99 (13.8)	280 (15.1)	0.038	98 (13.7)	168 (11.9)	0.053
CCI	4 [3, 5]	4 [3, 5]	0.002	4 [3, 5]	4 [3, 5]	0.019
ECOG performance						
0	588 (81.8)	1419 (76.6)	0.127	586 (81.7)	1148 (81.4)	
1	126 (17.5)	414 (22.4)	0.121	126 (17.6)	252 (17.9)	0.008
2	5 (0.7)	17 (0.9)	0.025	5 (0.7)	10 (0.7)	0.008
3	0 (0)	2 (0.1)	0.046	0 (0)	0 (0)	0.001
Operation type						
Laparoscopy	644 (89.6)	1481 (80.0)	0.27	643 (89.7)	1271 (90.1)	0.016
Robotic	1 (0.1)	188 (10.2)	0.465	1 (0.1)	2 (0.1)	0.001
Laparotomy	74 (10.3)	183 (9.9)	0.012	73 (10.2)	137 (9.7)	0.016
T staging						
T0	16 (2.2)	40 (2.2)		15 (2.1)	26 (1.8)	
T1	179 (24.9)	483 (26.1)	0.029	179 (25.0)	362 (25.7)	0.018
T2	238 (33.1)	566 (30.1)	0.055	238 (33.2)	465 (33.0)	0.005
T3	254 (35.3)	629 (34.0)	0.029	252 (35.3)	496 (35.2)	0.003
T4	32 (4.5)	134 (7.2)	0.119	32 (4.5)	61 (4.3)	0.007
Tumor location						
Right	218 (30.3)	504 (27.2)	0.069	217 (30.3)	445 (31.6)	0.028
Left	497 (69.1)	1327 (71.7)	0.054	496 (69.2)	957 (67.9)	0.029
Rectum	4 (0.6)	21 (1.1)	0.073	4 (0.6)	8 (0.6)	0.01
Lymphatic invasion	207 (28.8)	537 (29.0)	0.004	206 (28.7)	407 (28.9)	0.003
Perineural invasion	311 (43.3)	811 (43.8)	0.01	310 (43.2)	601 (42.6)	0.013
Vascular invasion	58 (8.1)	137 (7.4)	0.025	58 (8.1)	99 (7.0)	0.041
Anaesthesia time, min	173 [149, 204]	176 [143, 226]	0.115	173 [149, 203]	169.5 [141, 213]	0.014
Transfusion	9 (1.3)	43 (2.3)	0.081	9 (1.3)	18 (1.3)	0.002
Adjuvant therapy	445 (61.9)	1131 (61.1)	0.038	443 (61.8)	840 (59.6)	0.046

Values are expressed as mean (SD), median [IQR] or number (%). History of surgery included abdominal surgeries and cesarean section. Charlson Comorbidity Index (CCI) is a method to estimate 10-year survival in patients with multiple comorbidities. Heavy drinking, consuming 15 drinks or more per week for men or eight drinks or more per week for women as defined by the Centers for Disease Control and Prevention. Eastern Cooperative Oncology Group (ECOG) performance status: 0 Fully active; no performance restrictions, 1 Strenuous physical activity restricted; fully ambulatory and able to carry out light work, 2 Capable of all self-care but unable to carry out any work activities. Up and about >50% of waking hours, 3 Capable of only limited self-care; confined to bed or chair >50% of waking hours, 4 Completely disabled; cannot carry out any self-care; totally confined to bed or chair. ASA, American Society of Anesthesiologists; BMI, body mass index; CAD, coronary artery disease; CCI, Charlson Comorbidity Index, CKD, chronic kidney disease; COPD, chronic obstructive pulmonary disease; ECOG, Eastern Cooperative Oncology Group; NLR, Neutrophil/Lymphocyte ratio.

**Table 2 jcm-11-02648-t002:** Comparison of early postoperative complications with propofol- and sevoflurane-based anesthesia before and after propensity score matching.

	Overall Patients			After Matching		
	Propofol	Sevoflurane	*p*	Propofol	Sevoflurane	*p*
	(*n* = 719)	(*n* = 1852)		(*n* = 717)	(*n* = 1410)	
Morbidity	93 (12.9)	218 (11.8)	0.42	93 (13.0)	162 (11.5)	0.34
CD classification > grade III	15 (2.1)	42 (2.3)	0.88	15 (2.1)	28 (2.0)	0.84
Wound problems	6 (0.8)	16 (0.9)	1.0	6 (0.8)	11 (0.8)	0.87
Ileus	24 (3.3)	51 (2.8)	0.44	34 (3.4)	37 (2.6)	0.32
Anastomasis leakage	7 (1.0)	15 (0.8)	0.64	7 (1.0)	8 (0.6)	0.30
Intrabdominal fluid collection	1 (0.1)	12 (0.7)	0.13	1 (0.1)	12 (0.9)	0.002
Re-operation	4 (0.6)	6 (0.3)	0.48	4 (0.6)	3 (0.2)	0.20
Sepsis	1 (0.1)	0	0.28	1 (0.1)	0	<0.001
Myocardiac infarction	1 (0.1)	5 (0.3)	1.0	1 (0.1)	3 (0.2)	0.70
Pulmonary complication	2 (0.3)	1 (0.1)	0.19	2 (0.3)	1 (0.1)	0.22
Cerebral infarction	0	2 (0.1)	1.0	0	2 (0.1)	<0.001
Vascular complication	0	4 (0.2)	0.58	0	2 (0.1)	<0.001
Urinary retention	35 (4.9)	48 (2.6)	0.006	35 (4.9)	36 (2.6)	0.008

Values are expressed as number (%). In overall patients, the outcomes were compared with Chi-square test or Fisher’s exact test. In the propensity matched cohort, the risks of each outcome were compared with logistic regression using generalized estimating equations. Morbidity means the incidence of any complication. The Clavien-Dindo (CD) classification issued to evaluate the severity of surgical complications.; Grade III-V indicates major complications. The definition of complications was as follows: wound problem; occurrence of infection involving the skin or subcutaneous tissue and requiring surgical re-intervention, ileus; symptomatic and diagnosed by radiographic findings, anastomosis site leakage; diagnosed by radiographic findings, Intraabdominal fluid collection; diagnosed by radiographic findings, sepsis; despite adequate fluid resuscitation, patients have hypotension requiring vasopressors to maintain a mean arterial blood pressure above 65 mm Hg and have an elevated serum lactate concentration of more than 2 mmol/L resulting from dysregulated host responses to infection, myocardiac infarction; detection of a rise cardiac troponin values with symptoms, pulmonary complication; requiring treatment with antibiotics for a suspected respiratory infection or management by respiratory care physiotherapists for lung care, cerebral infarction; diagnosed by imagining findings, vascular complication; diagnosed by imagining findings, urinary retention; need for in-and-out catheterization or reinsertion of an indwelling urinary catheter during the hospital stay after the original urinary catheter had been removed.

**Table 3 jcm-11-02648-t003:** Cox regression proportional hazard overall survival and recurrence-free survival: multivariable model for overall patients.

	Recurrence-Free Survival		Overall Survival	
Variables	Hazard Ratio (95%CI)	*p*	Hazard Ratio (95%CI)	*p*
Propofol (ref. sevoflurane)	1.04 (0.80, 1.34)	0.80	1.12 (1.06, 1.17)	0.91
Age, year	1.02 (1.01, 1.03)	0.004	1.05 (1.03, 1.07)	0.000
ASA (ref. I)		0.025		0.61
II	1.11 (0.82, 1.50)	0.46	0.71 (0.41, 1.26)	0.24
III or IV	1.76 (1.14,2.73)	0.01	0.93 (0.42, 2.04)	0.85
Preoperative test				
Haemoglobin, mg/dL	1.00 (0.98, 1.01)	0.61	0.99 (0.95, 1.04)	0.68
Albumin	0.83 (0.65, 1.07)	0.15	0.14 (0.72, 0.47)	0.14
Creatinine, g/dL				
Neutrophil/Lymphocyte ratio	1.07 (1.02, 1.11)	0.006	1.12 (1.06, 1.17)	0.000
Neoadjuvant therapy	1.70 (1.23, 2.34)	0.001		
ECOG performance (ref. 0)		0.73		0.09
1	0.91 (0.70, 1.18)	0.48	1.27 (0.80, 2.01)	0.31
2	1.12 (0.51, 2.49)	0.78	1.29 (0.42, 4.00)	0.66
3	2.30 (0.30, 17.86)	0.43	22.84 (1.80, 290.72)	0.02
Operation (ref. laparotomy)		0.005		0.009
Laparoscopy	1.17 (0.72, 1.88)	0.53	0.51 (0.31, 0.84)	0.009
Robotic	1.66 (1.22, 2.25)	0.001	0.15 (0.24, 0.89)	0.04
T staging (ref. T0 or T1)		0.000		0.000
T2	1.24 (0.77, 2.00)	0.37	0.99 (0.38, 2.55)	0.98
T3	2.88 (1.79, 4.65)	0.000	2.32 (0.90, 5.98)	0.08
T4	88.36 (4.87, 14.35)	0.000	8.48 (2.95, 24.40)	0.000
Tumor location (ref. Right)				0.05
Left			0.60 (0.39, 0.92)	0.019
Rectum			1.51 (0.19, 11.92)	0.70
Lymphatic invasion	1.44 (1.12, 1.86)	0.005	1.62 (1.01, 2.60)	0.048
Perineural invasion	1.67 (1.27, 2.21)	0.000	1.42 (0.84, 2.37)	0.19
Vascular invasion	1.84 (1.38, 2.47)	0.000	0.40 (0.20, 0.55)	0.000
Anaesthesia time, min	1.00 (1.00, 1.00)	0.16	1.00 (1.00, 1.00)	0.18
Transfusion	0.93 (0.51, 1.71)	0.81	1.20 (0.50, 2.89)	0.69
Postoperative treatment	0.94 (0.67, 1.34)	0.74	0.68 (0.37, 1.25)	0.21

ASA, American Society of Anesthesiologists; BMI, body mass index; CAD, coronary artery disease; CCI, Charlson Comorbidity Index, CKD, chronic kidney disease; COPD, chronic obstructive pulmonary disease; ECOG, Eastern Cooperative Oncology Group.

## Data Availability

The data that support the findings of this study are available from the corresponding author upon reasonable request.

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
