# Peer review of "Early and Long-Term Outcomes after Propofol-and Sevoflurane-Based Anesthesia in Colorectal Cancer Surgery: A Retrospective Study"

_jcm, 2022, doi:10.3390/jcm11092648_

Round 1
Reviewer 1 Report
Suggestion:
Reconsider the text in order to decrease the plagiarism
The distribution of infections are equal between groups, because this is crucial for the results.
Author Response
Referee: 1
Comments to the Author
Reconsider the text in order to decrease the plagiarism
The distribution of infections are equal between groups, because this is crucial for the results.
- We thank the reviewer for this comment and have taken steps to further improve readability and flow. We have already checked the text by iThenticate, there was no problem with the plagiarism (<10%). We have revised with some re-arrangement the Discussion accordingly. (Page 11; in the discussion)
We agreed with reviewer’s comment. We added this point in the discussion. (Page 10, line 253)
Reviewer 2 Report
The article is difficult to read because there is an ambiguity that starts with the title which begins with the NLR.
Is this a new study on the influence of the mode of anesthesia on survival after colorectal surgery?
Is this a new study on the link between NLR and survival after colorectal surgery?
You have to choose?
Was it possible/useful to do a 4-group analysis using preoperative NLR? propofol and « normal range » of NLR vs propofol and « abnormal » NLR vs sevoflurane and « normal range » of NLR vs sevoflurane and « abnormal » NLR
Author Response
Referee: 2
Comments to the Author
The article is difficult to read because there is an ambiguity that starts with the title which begins with the NLR.
- Thank you for your constructive comments. As per the recommendation, we revised the title accordingly. “Early And Long-term Outcomes After Propofol- And Sevoflurane-based Anesthesia In Colorectal Cancer Surgery: A Retrospective Study” (Manuscript location: Page 1, title)
Is this a new study on the influence of the mode of anesthesia on survival after colorectal surgery?
Is this a new study on the link between NLR and survival after colorectal surgery?
You have to choose?
- We thank the reviewer for pointing this out. However, there was still paucity of data because of small sample size studies or limited report on intravenous anesthesia (TIVA) vs. inhalational anesthesia. Especially, the data in the anesthetic technique and postoperative NLR were little evidence. Thus, we designed this retrospective cohort study with NLR with clinical outcomes in the anesthetic technique. Despite TIVA seeming to lead to decreased mortality in patients with cancer in previous large retrospective studies, there is currently no evidence of high quality supporting TIVA being a superior choice compared with inhalational anesthesia. Previous studies have reported on the predictive potential of NLR as a prognostic factor in various types of malignancies, including resectable colon cancer. Resectable cancers of the colorectum, breast, prostate, and lung represent almost half of the overall cancer burden worldwide. Among those cancer surgery, consistent results (The anesthetic techniques were not significantly associated with cancer outcomes) were reported in breast cancer surgery. (Anesthesia and Circulating Tumor Cells in Primary Breast Cancer Patients; Anesthesiology 2020; 133:548–58 & Total Intravenous Anesthesia versus Inhalation Anesthesia for Breast Cancer Surgery; Anesthesiology 2019; 130:31–40 & Recurrence of breast cancer after regional or general anaesthesia: a randomised controlled trial; Lancet 2019; 394: 1807–15).
However, results in hematogenous metastasis cancer including colon cancer and anesthetic technique remain mixed. There is still no guidelines for anesthetic induction in cancer patients, thus, our study can be advised until further evidence is provided.
Was it possible/useful to do a 4-group analysis using preoperative NLR? propofol and « normal range » of NLR vs propofol and « abnormal » NLR vs sevoflurane and « normal range » of NLR vs sevoflurane and « abnormal » NLR
- We thank the reviewer for these insightful and helpful comments. Among the patients who showed normal NLR or high NLR, the comparison of propofol anesthesia and sevoflurane anesthesia may be possible and interesting. However, instead of that method, we choose the propensity score matching. Thus, we thought the association of anesthetic technique and preoperative NLR was not significant.
Also, previous studies (Neutrophils to lymphocytes ratio as a useful prognosticator for stage II colorectal cancer patients reported that pretreatment NLR over 3 may be a poor prognostic factor for tumor outcomes; BMC Cancer (2018) 18:1202 & Neutrophil-to-lymphocyte ratio is a prognostic factor for colon cancer: a propensity score analysis; BMC Cancer (2020) 20:922) were defined NLR at least above 3 might be related with significantly poor outcomes. However, our patients showed preoperative NLR (median [IQR] 2.1[1.5, 2.9]). Thus, the 4-group analysis using preoperative NLR may be not meaningful in current data. A larger and more balanced study sample size between the groups may be needed to clearly determine a subtle difference of preoperative NLR associated with anesthetic technique.
We thank the reviewer for this helpful comment. We have revised the Discussion with this point. (Manuscript location: Page 11, line 301-306)
Round 2
Reviewer 1 Report
I have no further comments for the authors
Author Response
Comments to the Author
I have no further comments for the authors
➢ Thank you very much for your prompt review and for the helpful comments about our manuscript.
Reviewer 2 Report
I stand by my original assessment. Changing the title does not make the article any better. The confusion remains: study of the influence of the mode of anesthesia on mortality? study of the influence of the mode of anesthesia on the ratio.
The authors emphasize the influence of the mode of anesthesia on mortality but the hypothesis described is: "We hypothesized that those patients receiving propofol anesthesia would have lower NLR compared to those undergoing sevoflurane anesthesia after colorectal surgery at the postoperative day 5."
Author Response
Comments to the Author
I stand by my original assessment. Changing the title does not make the article any better. The confusion remains: study of the influence of the mode of anesthesia on mortality? study of the influence of the mode of anesthesia on the ratio.
The authors emphasize the influence of the mode of anesthesia on mortality but the hypothesis described is: "We hypothesized that those patients receiving propofol anesthesia would have lower NLR compared to those undergoing sevoflurane anesthesia after colorectal surgery at the postoperative day 5.
- Thank you for your constructive comments. As the reviewer pointed out, too bread outcomes could lead to confusion about the main outcome. However, we noted that the primary goal of this study was NLR on the postoperative day as described in the introduction and method.
The followings are why the authors think the NLR at the postoperative day 5 as the primary outcome;
As we described in the previous response to reviewer’s comments, despite TIVA (intravenous anesthesia) seeming to lead to decreased mortality in patients with cancer in previous large retrospective studies, there is currently no evidence of high quality supporting TIVA being a superior choice compared with inhalational anesthesia. Thus, we tried to provide some further evidence with laboratory results. (Please see page 10 line 254-line 266) We hypothesized if anesthetic technique affects the immune response or systemic inflammatory response during early phase in patients who underwent surgical procedure, it can affect postoperative complications and recurrence/mortality. We thought the prognostic value of NLR in the immediate postoperative day might be limited, because surgical stress and wound healing have impact on inflammatory indicators. And our institution protocol have the examination of NLR at the postoperative day 5.
We could find the difference of postoperative NLR, however, there was no difference in the complications or recurrence/mortality between the two groups. The limitations of current study were the median follow-up (< 5 years), a single center study, and sample size.
As previous reviewer’s comments, we added the idea in the conclusion. “further study focusing on how the difference in postoperative NLR values according to the anesthetic technique revealed in this study affects the long-term prognosis is needed.”. (Page 11, line 332-334)
